# An Inverse Method to Determine Mechanical Parameters of Porcine Vitreous Bodies Based on the Indentation Test

**DOI:** 10.3390/bioengineering10060646

**Published:** 2023-05-26

**Authors:** Haicheng Zu, Kunya Zhang, Haixia Zhang, Xiuqing Qian

**Affiliations:** 1School of Biomedical Engineering, Capital Medical University, Beijing 100069, China; 2Beijing Key Laboratory of Fundamental Research on Biomechanics in Clinical Application, Capital Medical University, Beijing 100069, China

**Keywords:** vitreous body, liquefaction, indentation test, mechanical properties, inverse method

## Abstract

The vitreous body keeps the lens and retina in place and protects these tissues from physical insults. Existing studies have reported that the mechanical properties of vitreous body varied after liquefaction, suggesting mechanical properties could be effective parameters to identify vitreous liquefaction process. Thus, in this work, we aimed to propose a method to determine the mechanical properties of vitreous bodies. Fresh porcine eyes were divided into three groups, including the untreated group, the 24 h liquefaction group and the 48 h liquefaction group, which was injected collagenase and then kept for 24 h or 48 h. The indentation tests were carried out on the vitreous body in its natural location while the posterior segment of the eye was fixed in the container. A finite element model of a specimen undertaking indentation was constructed to simulate the indentation test with surface tension of vitreous body considered. Using the inverse method, the mechanical parameters of the vitreous body and the surface tension coefficient were determined. For the same parameter, values were highest in the untreated group, followed by the 24 h liquefaction group and the lowest in the 48 h liquefaction group. For C_10_ in the neo-Hookean model, the significant differences were found between the untreated group and liquefaction groups. This work quantified vitreous body mechanical properties successfully using inverse method, which provides a new method for identifying vitreous liquefactions related studies.

## 1. Introduction

The vitreous body is located between the lens and the retina, accounting for approximately two thirds volume of the eye globe. The normal vitreous body is composed of 98–99% water with a network comprising collagen fibers and hyaluronic acid (HA) [1]. Due to its soft and gel-like nature, the vitreous body serves as a mechanical damper for the eye [2], holding the lens and retina in place and protecting them from impacts and vibration [3]. The main reason for vitreous liquefaction with aging occurs due to the combination of collagen loss and vitreoretinal adhesion weakening [4,5]. Nowadays, vitreous liquefaction can be considered as the main reason of many ocular disorders, such as retinal tears, retinal detachment, vitreomacular traction, macular pucker, macular hole, etc. [6,7]. Existing studies have reported mechanical properties of vitreous body varied after liquefaction [8,9,10,11], suggesting that mechanical properties could be effective parameters to identify vitreous liquefaction process. Therefore, identifying vitreous liquefaction by mechanical properties can provide insights for studies on vitreous liquefaction related diseases.

There have been some studies using conventional methods to obtain mechanical properties of the vitreous body. A rheometer has been widely applied in studying viscoelastic properties of the vitreous body, where the storage and loss module or creep compliance can be determined [8,9,10,12,13,14,15,16,17,18,19,20,21]. In order to understand vitreous liquefaction, such studies were conducted on age-related changes of mechanical properties [9,10,17] and the enzymatic degradation of mechanical properties of the vitreous body [8,20]. In addition to rheometers, the mechanical testing machine was also used to determine the elastic module of the vitreous body using compression tests while it was put on a 35 mm diameter Petri dish [14]. However, all these vitreous body specimens were removed from the eyeball, which may lead to changes in the mechanical properties [22,23].

There are some studies on mechanical properties of vitreous bodies in its natural location. An optically trapped silica bead was used as a local probe to measure the micro-rheology of the vitreous body [24]. The elastic modulus and surface tension value of the local vitreous body in eye and in vitro were determined by a technology of cavitation induction [23]. The vitreous body with almost intact structures and surrounding tissues was tested by the rheometer with a self-devised probe [25]. Viscoelasticity of the vitreous body was measured by tracing intraocular microprobes while the magnetic force was applied [26]. Material anisotropy of the vitreous body in the posterior chamber could also be measured [27]. From the above studies, measurements were achieved with the aid of self-design testing systems, which were difficult to be used widely.

The indentation test is a very attractive method to obtain the mechanical properties of materials in a local area or at different length scales. It has been widely used to determine the mechanical properties of biological tissues [28,29,30,31,32,33,34,35,36], such as bone, liver, tooth, skin, cornea, iris, cell, etc. Elastic mechanical properties of some tissues, such as human skin or cornea, could be measured in vivo [28,33,34]. Therefore, we chose the indentation test to measure the mechanical properties of vitreous body in order to keep it its physiological conditions. Generally speaking, the elastic modulus of materials can be determined by the empirical formula and the force–depth curves in the indentation test. Recently, Chen [11] proposed a simple indentation method to identify vitreous liquefaction by comparing of relative differences of some typical mechanical parameters during relaxation; however, the pressure acting on the indenter was only recorded, which was probably affected by surface tension of the vitreous body. In fact, an elasto-capillary liquid bridge formed between the probe and the vitreous body droplet as the sample dripped from the probe to the rest of the droplet [37]. Forces due to surface tension were proved to affect elastic deformation, leading to the elasto-capillary phenomena [38]. The surface tension was calculated using cavitation rheology when the syringe needle was pulled upwards by the surface tension of the vitreous body upon insertion [23]. For these above reasons, the interaction caused by surface tension of the vitreous body should be taken into consideration during contact.

Due to the existence of surface tension of the vitreous bodies in the indentation test, the mechanical properties of the vitreous bodies could not be obtained by the empirical formula based on indentation test because the application conditions of the empirical formula are not satisfied. Due to this, an inverse finite element method is often used to determine the mechanical properties of materials, combining the finite element method (FEM) with the experimental results of indentation [34,39,40,41,42,43]. FEM is a numerical method used to solve the differential equations and has been widely used to analyze the mechanical response of complex organisms under the action of forces [44,45,46,47,48,49,50,51]. The mechanical parameters of the porcine vitreous body were estimated by combining the experimental results and finite element method [22].

In this study, the indentation tests were performed to obtain the indentation depths and indentation loads while the vitreous bodies were kept in its physiological conditions. Next, the finite element model was constructed to simulate the indentation test with the surface tension of the vitreous bodies considered. Finally, the inverse finite element method was used to determine the mechanical parameters of the untreated and liquefied vitreous bodies, which could be used to distinguish the untreated and liquefied vitreous bodies.

## 2. Materials and Methods

### 2.1. Indentation Tests

Porcine eyes were harvested from a local slaughterhouse and transported to the lab within 4 h after enucleation. All the eyeballs were cleaned by removing the excessive tissues such as extraocular muscles and fat. They were randomly divided into three groups including the untreated group, the 24 h liquefaction group and the 48 h liquefaction group. There were seven specimens in every group. For liquefaction groups, as enzymatic breakdown of the collagen fibrils probably causes age-related changes in the vitreous body [52], 0.1 mL collagenase (0.5 mg/mL) was injected through the pars plana to the vitreous body chamber by a 1 mL syringe to induce liquefaction of the vitreous body [8,20]. Then, specimens in the 24 h liquefaction group and in the 48 h liquefaction group were preserved in a humidity box for 24 h and 48 h, respectively before the indentation test.

Before the indentation test, the specimen was fixed. Firstly, the eyeball was placed in a container, whose inner wall is a half spherical surface with a radius of 5 mm. The cornea was exposed outside the container. Then, the eyeball was adjusted until the axes of the eyeball and the container coincided, the eyeball was glued to the inner wall of the container. Finally, the cornea and part of the sclera that spilled outside the container were removed with a sharp knife, and the exposed vitreous body was used for indentation experiments.

The indentation tests were conducted using the mechanical testing machine (ElectroForce 3100, Bose Corp., Eden Prairie, MN, USA) with a calibrated high-precision force sensor whose capacity is 225 g (Honeywell 31, Honeywell, Columbus, OH, USA), as shown in Figure 1.

The indentation load might be affected by the indentation speed of the indenter. In order to investigate the impact of the speed on the indentation load—depth curves, five different speeds were adopted in the study including 0.04, 0.08, 0.16, 0.32 and 0.64 mm/s. Five specimens were adopted in the indentation tests. The results showed that there were no obvious differences among specimens at different speeds (Figure 2). Therefore, we set the indentation speed at 0.04 mm/s.

Before the indentation test, the container was fixed with the clamp of the machine, a spherical indenter with a radius of 5 mm was adjusted to ensure there was a small gap between the indenter and the upper surface of vitreous body. Then, the indenter moved downwardly, controlled by the mechanical testing machine. When the initial contact between the indenter and the surface of vitreous body happened, it was considered as the starting point of the test. The vitreous body was indented with the spherical indenter at the rate of 0.04 mm/s, meanwhile, the indentation load and the indentation depth were recorded until the indentation depth reached 3.8 mm.

### 2.2. Construction of the Finite Element Model

The axisymmetric finite element model was constructed with the finite element software ABAQUS (Version 2019, Simulia, Vélizy-villacoublay, France). ABAQUS has been widely used to analyze the mechanical response of biological tissues or organisms under the action of forces mentioned above [44,45,46,47]. In the finite element model, three main segments (i.e., indenter, vitreous body and eyeball wall were involved (Figure 3a). The spherical indenter was simplified as a rigid body because the mechanical properties of indenter far outweigh the vitreous body. The eyeball wall was simplified as a homogeneous tissue, and the thickness of the eyeball wall was taken as 0.5 mm thick [53]. As the sclera was the primary load-bearing structure of eyeball [54,55], we thought that the mechanical properties of the eyeball wall would be similar to those of the sclera. The eyeball wall was simplified as a linear elastic material with an elastic modulus of 3.0 MPa [56,57] and the Poisson’s ratio of 0.49 [58]. Based on the dimensions of the container where the specimen was placed during the intention test, the geometry of the specimen was constructed as a hemisphere with a radius of 12.0 mm.

As hydrogels were treated as neo-Hookean materials in most studies [59,60,61,62], in this study, the gel-like vitreous body was regarded as a neo-Hookean model governed by the strain energy function [63,64]:(1)$$U=C_{10}({\bar{I}}_{1}-3)+\frac{1}{D_{1}}(J^{el}-1{)}^{2}$$
where U is the strain energy density in the undeformed configuration and *C*_10_ and *D*_1_ are the mechanical parameters. *C*_10_ is half of the initial shear elastic modulus. *D*_1_ is a parameter indicating the compressibility of materials.

Surface tension occurs when the surface of a liquid is in contact with another phase, which is an important factor in the behavior of fluids. Research has shown that the surface tension would affect elastic deformation of a soft solid with fluid-filled droplet inclusions [38]. Given that a normal vitreous body is composed of 98–99% water with a network comprising collagen fibers and HA, the surface tension between the vitreous body and the indenter was taken into consideration in the model. In reference to [23], the surface tension of the vitreous body was defined as follows:(2)$$F_{\alpha }=2\pi \alpha r$$
where *F_α_* is the surface tension, *α* is the surface tension coefficient for the vitreous body, and *r* is the contact radius. *F_α_* is a vector whose direction is perpendicular to the contact surface between the vitreous body and the indenter, so the vertical component *F_α_*_1_ of the surface tension could be expressed as follows:(3)$$F_{\alpha 1}=\frac{2\pi \alpha h(2R-h)}{R}$$
where *R* is the radius of the indenter, and *h* is the indentation depth.

The contact between the spherical indenter and the upper surface of the vitreous body was simplified as frictionless contact [65,66]. The symmetrical constraint of the model along the axis were set. The vitreous body was tied with the inner surface of the eyeball wall [49] while its outer surface was fixed.

The model was meshed with two types of elements, which were the 4-node bilinear axisymmetric quadrilateral element and 3-node linear axisymmetric triangle element, as shown in Figure 3b,c. The wall of the eyeball included 648 nodes and 560 elements. For the vitreous body, the mesh convergence study was conducted to determine how the mesh density affect the accuracy of stress and strain prediction of the vitreous body. When the number of the elements increased from 6000 to 12,000, the results showed the von Mises stress was from 197.8 Pa to 197.7 Pa, and the maximum principal strain was from 0.2033 to 0.2034. As a result, the model employed 12,000 meshes and 12,151 nodes was used in the model.

### 2.3. The Inverse Method to Determine the Mechanical Properties

In order to improve the efficiency to obtain the finite element simulation curves that match the experimental indentation curves in inverse finite element method, we adopted optimization algorithms. There are many optimization algorithms [67,68,69], such as genetic algorithm, simulated annealing algorithm, ant colony algorithm, etc. In this study, the multi-island genetic algorithm (MIGA) [67,70,71] was conducted for the optimization. MIGA is one of the improved genetic algorithms. The entire optimization population is divided into several islands in MIGA, individuals in each island are selected, crossed, and mutated. The first-class individuals from each island regularly migrate to other islands. Finally, by iteration to the maximum generation, the optimal solution is achieved. MIGA improves efficiency and reliability by preventing the algorithm from falling into the local optimal solution in advance [72], quantified by the following objective function:(4)$$\varepsilon ={\sum_{d=1}^{D}\left(f^{exp}-f^{sim}\right)}^{2}$$
where $f^{exp},f^{sim}$ are the experimental and predicted loads exerted on the indenter, respectively, and *D* is the total number of indentation depth levels; in this study, 20 indentation depth levels were adopted.

The parameters of the MIGA were shown as follows: the number of islands was 10, number of generations was 10, size of populations was 10, crossover rate was 1, mutation rate was 0.01, migration rate was 0.01 and migration interval was 5.

As optimization algorithms, including the MIGA algorithm, are sensitive to ranges of the parameters to be identified, the surface tension coefficient α ranged from 0.035 to 0.24 N/m in reference to [23], while C_10_ was set from 10 to 300 Pa and D_1_ from 1 × 10^−4^ to 2 × 10^−2^ Pa^−1^ in reference to [9,23].

The process for determining the mechanical properties is shown Figure 4.

### 2.4. Statistical Analysis

In order to compare the differences of the mechanical parameters among different groups, statistical analyses were implemented using SPSS software (version 25, IBM. Inc., Armonk, NY, USA). One-way ANOVA was used to compare mechanical parameters in the different groups while a cutoff *p* value of 0.05 was used for indicating statistical significance.

## 3. Results

### 3.1. Experimental Results

The results suggest that the indentation load varied as the indention depth increases, which is illustrated in Figure 5. The specimens in different groups shared the same trends. At the beginning, as the indentation depth increased, the indentation load decreased and became negative. When the indenter depth reached about 0.5 mm, the indentation load dropped to the minimum values. Then, the indentation load increased to zero, ultimately to the maximum. The maximum indentation loads were highest in the untreated group, followed by the 24 h liquefaction group and were the lowest in the 48 h liquefaction group.

### 3.2. Identification Results of the Mechanical Parameters

Based on the indentation tests, mechanical parameters were determined using the inverse method, including *C*_10_ and *D*_1_, in the neo-Hookean model and the surface tension coefficient *α*. All the three parameters were identified in different groups, listed in Table 1. Whether the parameters *C*_10_ and *D*_1_ in the neo-Hookean model or the surface tension coefficient *α*, the values were the highest in the untreated group, the second highest in the 24 h liquefaction group and the lowest in the 48 h liquefaction group.

Figure 6 shows comparison between the experimental results and simulated results of the indentation loads. Simulated results were calculated using the identified mechanical properties of the untreated, 24 h liquefaction and 48 h liquefaction groups. The simulated results matched the experimental results well in each group, indicating that the identified values of *C*_10_, *D*_1_ and *α* could be used to characterize the mechanical behaviors of the vitreous body.

Identified mechanical parameters of different groups were compared, as shown in Figure 7. Statistical differences of *D*_1_ were not found among different groups. For *C*_10_ and *α*, values from the untreated group are significantly different from those from the 24 h liquefaction group and 48 h liquefaction group.

## 4. Discussion and Conclusions

The vitreous body occupies about two thirds of the eye volume, which plays an important role in maintaining the homeostasis of the eyeball and its manifold interactions with neighboring structures [10]. Vitreous liquefaction leads to phase separation and gel network collapse, causing complications, such as retinal detachment, macular holes, vitreous hemorrhage, and vitreous floaters [73]. In this study, we proposed an inverse method based on the indentation test to determine the mechanical properties of the vitreous body, which could be used to make a distinction between the untreated and liquefied vitreous body.

The indentation tests are often used to measure the mechanical behavior of the specimen that cannot be conveniently measured using a traditional tensile or compressive test. The vitreous body is fragile and easily deformable. The preparation of a traditional experimental specimen requires extraction of the vitreous body from the eyeball, which could lead to changes in the mechanical properties [22]. In order to ensure the specimen is as close to its original physiological conditions, we kept the ocular posterior segment fixed with the vitreous body in its natural location.

We have estimated the sample size based on the power according to values of C_10_ among different groups using one-way analysis of variance F-tests. The results showed that when the value of power was set as 0.85, the sample size was estimated to be seven. So it was reasonable to enroll seven specimens in each group in our study. In addition, the surface tension coefficient α showed significant differences between the untreated group and liquefaction groups through statistical analysis. We also estimated the sample size based on α among different groups. The sample size was estimated to be nine for each group if the value of power was set as 0.85. In a future work, we will enlarge the sample size to determine the statistical differences of α.

The surface tension of the vitreous body is important to understand the mechanical properties of the vitreous body. The surface tension between the vitreous body and the indenter were considered during the indentation test. Therefore, the indentation load was composed by two forces on the indenter (Figure 3). One was surface tension, and the other was the force applied by the mechanical testing machine. In the initial segment of the load–depth curves, the surface tension was higher than the force applied by the mechanical testing machine, resulting in presenting ‘tensile’ forces in the mechanical test. As the force applied by the mechanical testing machine increased, the indention load increased to zero, which meant the force applied by the mechanical testing machine was equal to the surface tension. Finally, the indentation load was kept positive, suggesting that the force applied by the mechanical testing machine was higher than the surface tension. Therefore, the surface tension of the vitreous body could not be ignored, which was approved by previous studies [8,11,23,37,38]. The adhesive force of the vitreous body was measured by stretching the vitreous body between the cleated parallel plate and stage of the rheometer [8]. Due to surface tension, an elasto-capillary liquid bridge formed between the probe and the vitreous body droplet as the sample dripped from the probe to the rest of the droplet [37]. The pressure acting on the indenter below zero during relaxation was probably affected by surface tension of the vitreous body [11]. The syringe needle was pulled upwards by the surface tension of the vitreous body when it was inserted in the vitreous body [23]. Surface tension of the vitreous body affected the pressure acting on the indenter during relaxation [11]. Surface tension of the vitreous body pulled the needle tip upwards when the syringe needle was inserted in the vitreous body [23]. Forces due to surface tension could lead to the elasto-capillary phenomena based on the finite element analysis of elastic deformation of a soft solid [38].

The inverse method is an effective way to determine mechanical properties and has been used to determine the mechanical properties of tissues [28,29,30,31,32,33,34,35,36,70,71]. In this study, mechanical parameters of vitreous body (i.e., *C*_10_, *D*_1_ in the neo-Hookean model and the surface tension coefficient *α*) were identified considering the surface tension. The results indicated that the highest values of *C*_10_ were 31.57 ± 6.91 Pa in the untreated group, followed by a decrease in values to 23.89 ± 4.95 Pa for the 24 h liquefaction group and the lowest values of 17.65 ± 4.35 Pa were found in the 48 h liquefaction group. *C*_10_ in the neo-Hookean model is related to the initial shear modulus, whose values decreased alongside the liquefaction process. Significant differences in *C*_10_ could also be found between the untreated group and 24 h liquefaction group (*p* = 0.018), while significant difference also existed between the untreated group and 48 h liquefaction group (*p* = 0.001). Significant differences in *C*_10_ could also be found between the 24 h liquefaction group and 48 h liquefaction group (*p* = 0.049). Our results agreed with the previous studies [8,20]. The network of collagen fibrils in the vitreous body was recognized as serving a load-bearing function [3]. When collagenase was injected to the vitreous body, the fibrillar backbone of the vitreous body was digested [8], then broken down [74]. The structure was easier to elastically deform due to fewer bonds present in the network of collagen [25]. Additionally, the values of *C*_10_ in the 48 h liquefaction group were the lowest, probably because the natural degradation of the vitreous bodies that occurred after the eyeball was removed for over 24 h [25], which aggravated the liquefaction of the vitreous bodies.

For the parameters *D*_1_, though no differences were found among different groups, the values changed among different groups as well as the parameter *C*_10_, the highest values were in the untreated group, the second highest ones in the 24 h liquefaction group and the lowest ones were in the 48 h liquefaction group. *D*_1_ is a parameter indicating the compressibility of materials. After liquefaction, values of *D*_1_ decreased because the bonds in the collagen network broke due to the injection of the collagenase to the vitreous body.

The surface tension coefficient was 0.0950 ± 0.0296 N/m in the untreated group. As the initial shear elastic modulus *G*_0_ = 2*C*_10_ in the neo-Hookean model, *G*_0_ was 63.14 ± 13.82 Pa in the untreated group. In the previous study [23], the elastic moduli of vitreous body in eye and in vitro were estimated to be 660 Pa and 120 Pa, respectively. The surface tension coefficients in eye and in vitro were 0.14 and 0.07 N/m, respectively. By contrast, the surface tension coefficient in the untreated group was higher than that in vitro, while it was lower than that in eye. This is probably because the vitreous body was measured in different locations, the upper surface of the vitreous body in our study versus the site of the needle tip inserted to the vitreous body. With regard to the moduli, if Poisson’s ratio was taken as 0.5, the initial elastic modulus would have been 189.42 ± 41.46 Pa, which were of the same order of magnitude as those [23] despite differences in methodology.

It is important for the inverse method to set the initial ranges of the parameters. We set the same range to identify the mechanical parameters in different groups for the same parameter.

There are some limitations in our study. In the model, the vitreous body was taken as homogenous. In fact, the gelatinous structure of the vitreous body did not have a homogenous density [75], and there existed difference in the rheological properties among different regions [12,13]. Moreover, vitreous liquefaction resulted in the formation of liquid-filled cavities, which could also lead to further heterogeneity of the vitreous body; however, it is worth mentioning that significant differences of the identified parameters were found between the untreated group and 24 h liquefaction group. The significant differences were also found between the untreated group and 48 h liquefaction group. It suggested that the mechanical properties of the untreated vitreous body was significantly different from that of the liquefaction groups. It indicated that it is feasible to distinguish the liquefied vitreous body from the untreated one using the mechanical properties identified based on the indentation test in this study. The untreated vitreous body exhibits heterogeneity, while the heterogeneity increases in the liquefied vitreous body due to liquefaction. In a future work, the finite element model of the vitreous body with an inhomogeneous structure or liquid-filled cavities will be constructed to estimate the local mechanical performance of the vitreous body. In addition, the vitreous body was taken as a hyper-elastic solid, despite being a gel-like material that acted as a damper to protect the eye. The indentation tests were performed on the same specimen at different speeds and the results showed that the curves of indentation load versus the indentation depth were similar, which suggested that the indentation speed had little influence on the results while indentation speed was smaller than 0.64 mm/s. In the future work, it is necessary to model the vitreous body as the viscoelastic solid to determine the effect of the viscoelasticity on its mechanical response. Finally, the vitreous body of porcine eye was used in this study due to difficulty in obtaining human eyes, though many studies adopted the porcine eye to understand the mechanical behavior of the vitreous body in its physiological conditions [22,24,25]. Differences of the mechanical behavior between the human eye and the porcine eye should be focused in the future work.

In conclusion, it is feasible to identify liquefaction of the vitreous body by the difference of mechanical properties of untreated and liquefied vitreous bodies, which were obtained by inverse finite element method based on the indentation test. Our study provides an effective approach to distinguish vitreous body liquefying process, which can deepen our understanding of vitreous liquefaction.

## Figures and Tables

**Figure 1 bioengineering-10-00646-f001:**
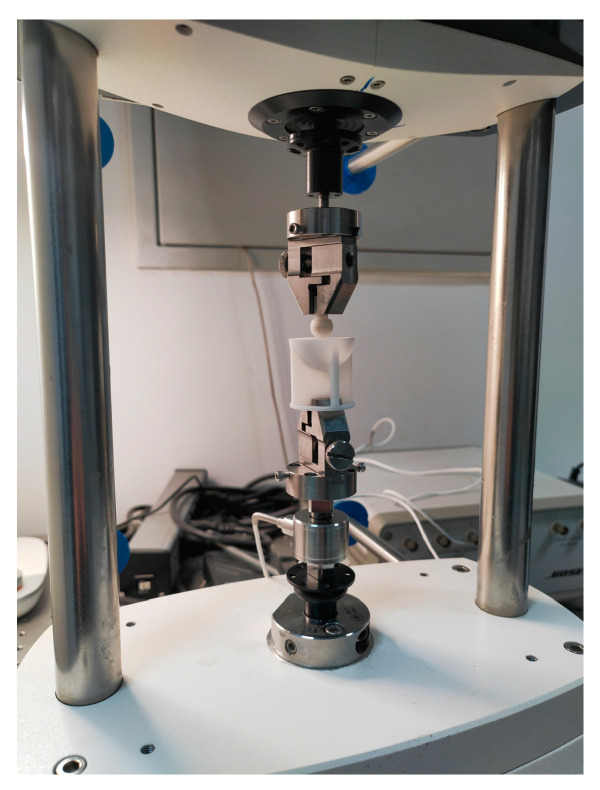
The overall experimental setup.

**Figure 2 bioengineering-10-00646-f002:**
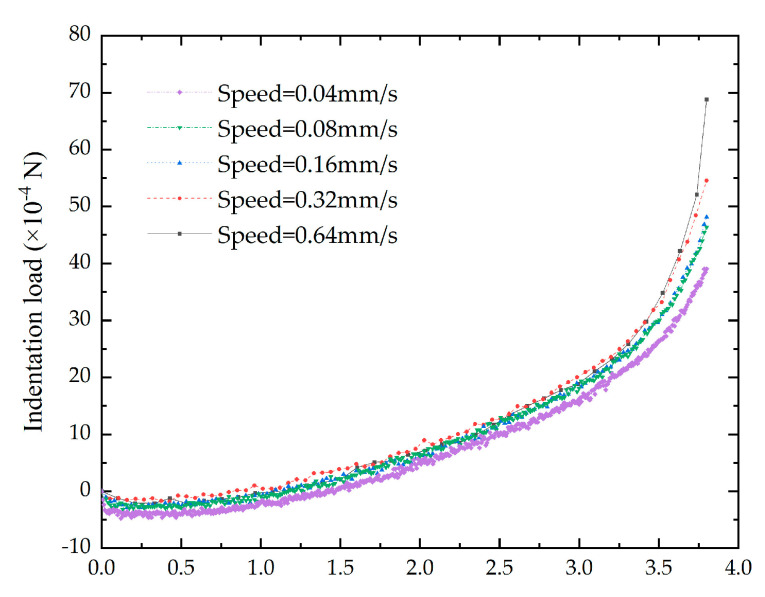
The effect of indentation speeds on the indentation load–depth curves.

**Figure 3 bioengineering-10-00646-f003:**
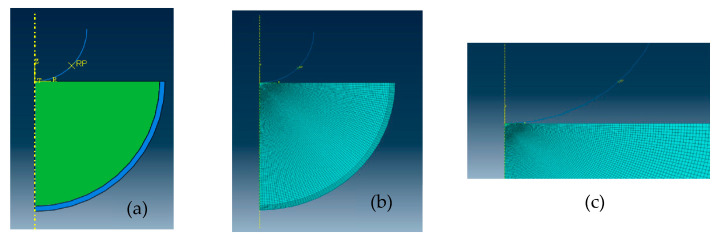
The finite element model. (**a**) The whole model; (**b**) meshes of the whole model; (**c**) meshes in the contact area between the indenter and the upper surface of the vitreous body.

**Figure 4 bioengineering-10-00646-f004:**
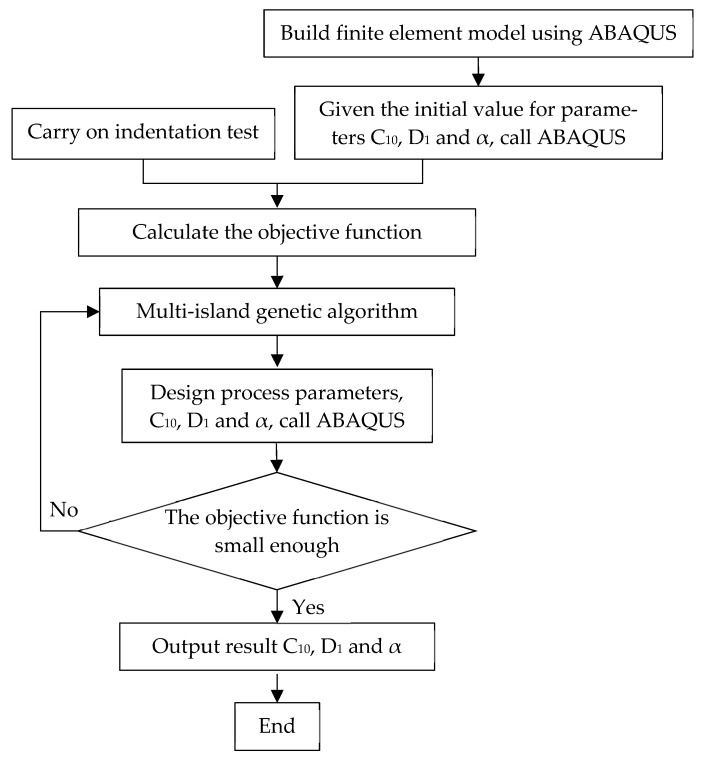
Process for determining the mechanical properties.

**Figure 5 bioengineering-10-00646-f005:**
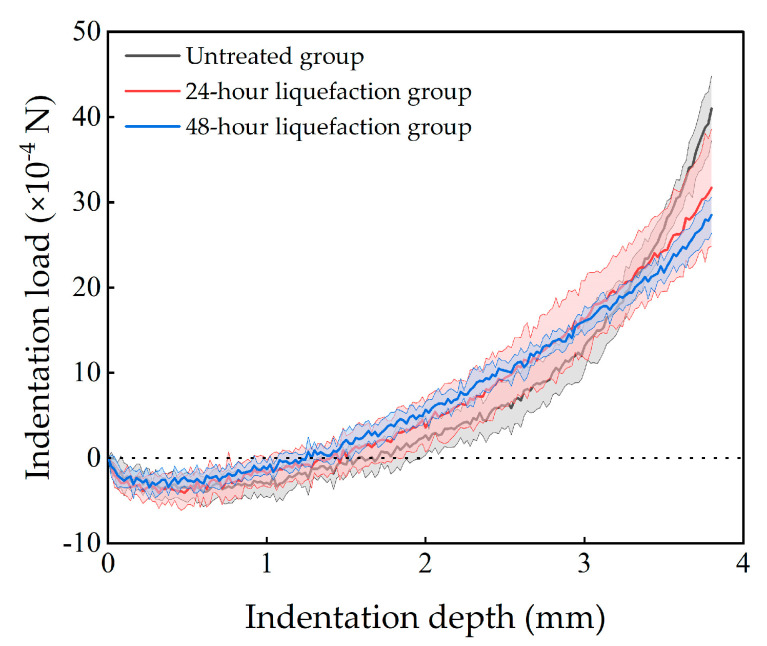
Average results of the indentation tests from different groups. The solid lines represent the averaged results, while the light−colored areas represent for corresponding standard deviation ranges.

**Figure 6 bioengineering-10-00646-f006:**
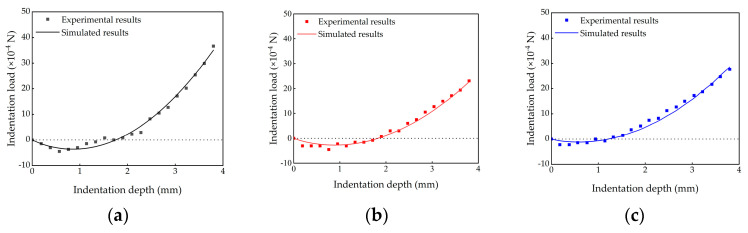
Comparison of experimental results and simulated results of the indentation loads using the identified mechanical properties of different groups. (**a**) Untreated group; (**b**) 24 h liquefaction group; (**c**) 48 h liquefaction group.

**Figure 7 bioengineering-10-00646-f007:**
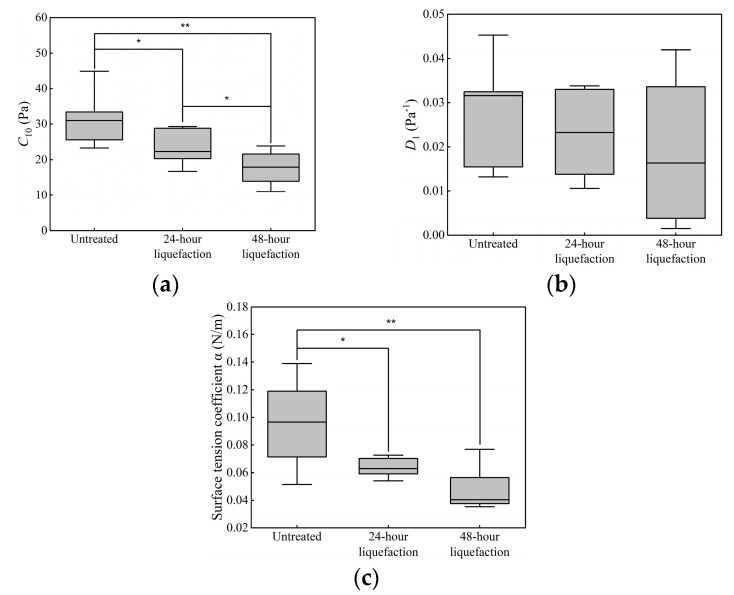
Results of identified mechanical parameters in different groups. (**a**) *C*_10_ of neo−Hookean model; (**b**) *D*_1_ of neo−Hookean model; (**c**) the surface tension coefficient α. * means *p* < 0.05, while ** means *p* < 0.01.

**Table 1 bioengineering-10-00646-t001:** Identified results of the mechanical parameters for the different groups.

Group	*C*_10_ (Pa)	*D*_1_ (×10^−2^ Pa^−1^)	α (N/m)
Untreated group	31.57 ± 6.91	2.79 ± 1.10	0.095 ± 0.0296
24 h liquefaction group	23.89 ± 4.95	2.33 ± 0.96	0.0632 ± 0.00647
48 h liquefaction group	17.65 ± 4.35	1.93 ± 1.53	0.0466 ± 0.0150

## Data Availability

Not applicable.

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
