# Peer review of "An Inverse Method to Determine Mechanical Parameters of Porcine Vitreous Bodies Based on the Indentation Test"

_bioengineering, 2023, doi:10.3390/bioengineering10060646_

Round 1

Reviewer 1 Report

The paper is in general very hard to understand, partly because of its bad English, and partly because of lack of explantations, especially regarding the methods. There are several mistakes, for example, what is aqueous body? Does it refer to vitreous? The method section is overly complicated, and I do not understand most of what they want to do and, more importantly, why they chose to do so, e.g., why they use the infinite model’ and ‘multi-island genetic algorithm’. It should be described what are these methods exactly and how they fit into the current work and why they were chosen. Furthermore, it is difficult to understand what the relevance of this work is and what do the results tell us. The paper should be re-written so that it is understandable for a ‘general’ scientist or medical doctor.

Must be significantly improved.

Author Response

All authors appreciate reviewer for the meaningful and useful comments. These comments are helpful for revising and improving the manuscript. The revisions in the manuscript and the responds to reviewer’s comments are shown in attachment.

Reviewer 2 Report

1. Authors used the word aqueous to refer to vitreous fluid quite interchangeably. In order to prevent any confusion for your readers please avoid using the word aqueous when you are really referring to the vitreous substance. Many will confuse the aqueous with the anterior chamber aqueous fluid.

2. In the abstract for the first time the mentioned abbreviations need to be explained.

3. Please avoid re-writing the result in the discussion section. Reduce the redundancy

4. Another limitation of your study is the sample size. One need to justify he sample size of each group based on the power of the study.

Author Response

(The authors gave the same response as above.)
